# Traffic Impact Area Detection and Spatiotemporal Influence Assessment for Disaster Reduction Based on Social Media: A Case Study of the 2018 Beijing Rainstorm

**Tengfei Yang [1,2], Jibo Xie [1,*], Guoqing Li [1], Naixia Mou [3], Cuiju Chen [3], Jing Zhao [1,2], Zhan Liu [1,4] and Zhenyu Lin [1,4]**

[1] Aerospace Information Research Institute, Chinese Academy of Sciences, Beijing 100094, China; yangtf@radi.ac.cn (T.Y.); ligq@radi.ac.cn (G.L.); zhao.j@aircas.ac.cn (J.Z.); 211704020001@home.hpu.edu.cn (Z.L.); 211704010005@home.hpu.edu.cn (Z.L.)

[2] School of Electronic, Electrical and Communication Engineering, University of Chinese Academy of Sciences, Beijing 100049, China

[3] College of Geomatics, Shandong University of Science and Technology, Qingdao 266590, China; mounx@lreis.ac.cn (N.M.); ccjccj94@163.com (C.C.)

[4] School of Surveying and Land Information Engineering, Henan Polytechnic University, Jiaozuo 454001, China

* Correspondence: xiejb@radi.ac.cn; Tel.: +86-1082178066

**Abstract:** The abnormal change in the global climate has increased the chance of urban rainstorm disasters, which greatly threatens people's daily lives, especially public travel. Timely and effective disaster data sources and analysis methods are essential for disaster reduction. With the popularity of mobile devices and the development of network facilities, social media has attracted widespread attention as a new source of disaster data. The characteristics of rich disaster information, near real-time transmission channels, and low-cost data production have been favored by many researchers. These researchers have used different methods to study disaster reduction based on the different dimensions of information contained in social media, including time, location and content. However, current research is not sufficient and rarely combines specific road condition information with public emotional information to detect traffic impact areas and assess the spatiotemporal influence of these areas. Thus, in this paper, we used various methods, including natural language processing and deep learning, to extract the fine-grained road condition information and public emotional information contained in social media text to comprehensively detect and analyze traffic impact areas during a rainstorm disaster. Furthermore, we proposed a model to evaluate the spatiotemporal influence of these detected traffic impact areas. The heavy rainstorm event in Beijing, China, in 2018 was selected as a case study to verify the validity of the disaster reduction method proposed in this paper.

**Keywords:** social media; traffic impact area detection; spatiotemporal influence assessment; disaster reduction

## 1. Introduction

In recent years, the abnormal change in the global climate has induced frequent meteorological disasters, especially torrential rainstorms in urban areas. This causes many problems for normal urban management and public travel. Although many modern monitoring methods are very helpful for the timely acquisition of disaster information, there are many limitations. For example, cameras distributed in urban areas can provide detailed disaster information of local areas in near real time,

but the cost of using these devices is high, and it is difficult to popularize them in some underdeveloped areas. Advanced radar remote sensing technology can overcome the influence of weather, which is fatal to traditional optical remote sensing [1,2]. However, the undulating buildings in the urban environment have a great impact on its imaging, and the longer revisit period is not conducive to the continuous observations of disasters.

Social media, as a new data source, has been widely used for disaster reduction. This data source is uploaded by the public spontaneously, which is similar to crowdsourcing data [3] and VGI [4] (volunteered geographic information). Social media has the advantages of rapid timeliness, good communication, and low acquisition cost. The rich disaster-related information contained in it (such as time, location, content, and network) is very helpful to understand the progress of disasters and gain disaster situation awareness. Recognizing the potential in disaster reduction, UNISDR (United Nations International Strategy for Disaster Reduction) adopted social media into the mainstream emergency communications under the Sendai Framework for Disaster Risk Reduction 2015 to 2030 on 15 March, 2015 [5]. However, the unstructured form of the data makes it difficult to use the disaster-related information efficiently. Thus, many researchers have tried to construct different methods or computing frameworks to process these data from different perspectives to serve disaster reduction. For example, Chae [6] proposed a centralized system that uses the spatiotemporal information contained in social media data to study public trajectory changes to assist the government in making disaster reduction decisions. In combination with other multi-source data, J Fohringer [7] and Li [8] proposed a method for the rapid mapping of flood ranges based on social media spatiotemporal information. Based on a Bayesian model, Sakaki [9] proposed a framework that can extract disaster-related topics from social media text and then combine it with spatiotemporal information contained in social media to detect the occurrence of disasters. Some other researchers have considered extracting public emotional information contained in social media to study disaster progression [10-12]. This information can provide insight into the disaster situation, rescue effect, etc., from the public's subjective feelings. In addition, compared to other data, social media data contain unique interactive information that is often expressed through operations such as praise and forwarding. These interaction records and patterns provide great opportunity to investigate information dissemination, exchange, update, etc., in various agents (e.g., ordinary users, authoritative agencies, and news media) [13]. Many researchers have taken this information to study different disasters, such as floods[14], hurricanes [15], and riots [16], which reflect more knowledge about individuals, groups, social phenomena, etc. Although the current research on using social media for disaster reduction is very comprehensive, there are still some shortcomings, such as the coarse granularity of disaster information mining and rare comprehensive analysis of multi-dimensional disaster-related information. For example, when urban rainstorms occur, how to quickly extract detailed disaster-related information to help detect traffic impact areas, how to feedback the severity of the affected areas based on public emotions, and how to assess the spatiotemporal influence of disasters are important questions. Thus, in this paper, we took a rainstorm event in Beijing on 16 July, 2018, as a case to address these problems from the following two aspects.

### 1.1. Extracting Multi-Dimensional Disaster-Related Information to Help Detect Traffic Impact Areas

Many researchers have used text mining methods to obtain disaster information. For example, Laylavi [17] took term, word frequency, and event correlation to extract storm disaster events from social media data to assist in monitoring disasters. Fang [18] and Wang [19] applied an unsupervised learning model to classify disaster themes contained in social media text to study public attention in rainstorm disasters. Through model calculations, some disaster information related to traffic impact can be identified. Other researchers have considered taking natural language processing methods to extract urban waterlogging information to assist in analyzing traffic impacts [20,21]. However, these existing methods are coarse-grained in extracting disaster information. They rarely focused on detailed road condition information, which can reflect the specific impact on traffic. In addition, existing methods have also rarely considered the public emotional information contained in social media to comprehensively analyze the traffic impact. On the one hand, public emotional information

reflects the public's subjective feelings about the disaster, public requirement, rescue effect, etc., which are difficult to collect from other disaster monitoring data. On the other hand, not every micro-blog contains obvious road condition information. Thus, public emotion can be considered effective supplementary information. In this paper, we integrate text mining algorithms, including natural language processing, semantic knowledge base, and deep learning, to extract fine-grained road condition information and public emotion information contained in social media texts. Furthermore, we combine this information with time, location and road network data to detect the traffic impact areas. Analysis results provide more effective information for disaster reduction, greatly improving the rescue efficiency.

## 1.2. Spatiotemporal Influence Assessment of Disasters

Social media contains multiple interaction patterns, based on which many researchers have been mining more disaster knowledge. However, few authors have considered introducing geographic location information into these interaction patterns. This is because the common interaction operations in social media, including forwarding, likes, etc., are usually not geo-tagged. Some researchers [22,23] have attempted to overcome this limitation of social media by using social media user profile locations to replace the user's actual location, which are very applicable for understanding the spatial dissemination of information and the spatial influence of disaster events. However, these methods still have some disadvantages. On the one hand, it is difficult to further understand the spatial impact distribution of disaster events on a small spatial scale (user profile locations are mostly at the city level). On the other hand, existing information interaction research has rarely considered public emotional information, and different emotional information reflects the different influence degree of disaster events. In this paper, we would like to further understand the magnitude and distribution of the spatial influence of traffic impact areas. Thus, we constructed a spatial influence assessment model based on the co-reference relationship of these areas. That is, when micro-blogs in different locations talk about a certain traffic impact area, they will interact with this area. Furthermore, we refer to the modeling method of complex networks [24-26] and combine it with the public emotional information to build evaluation indicators, including attention degree, interaction degree, and weighting degree, to quantify the spatial influence of traffic impact areas. These indicators are related not only to the number of relevant micro-blogs but also to the emotional information contained in micro-blogs. Because different emotion categories reflect different reactions of the public to disaster , areas with more negative emotions may imply greater impacts.

## 2. Methodology

In this paper, we propose a framework that can automatically acquire, parse, and process social media data and then intelligently mine disaster-related information contained in it, including time, location, fine-grained road condition information, and public emotional information. Finally, the framework provides effective information for disaster reduction through detecting traffic impact areas and assessing the spatiotemporal influence of disasters. The overall framework structure is shown in Figure 1.

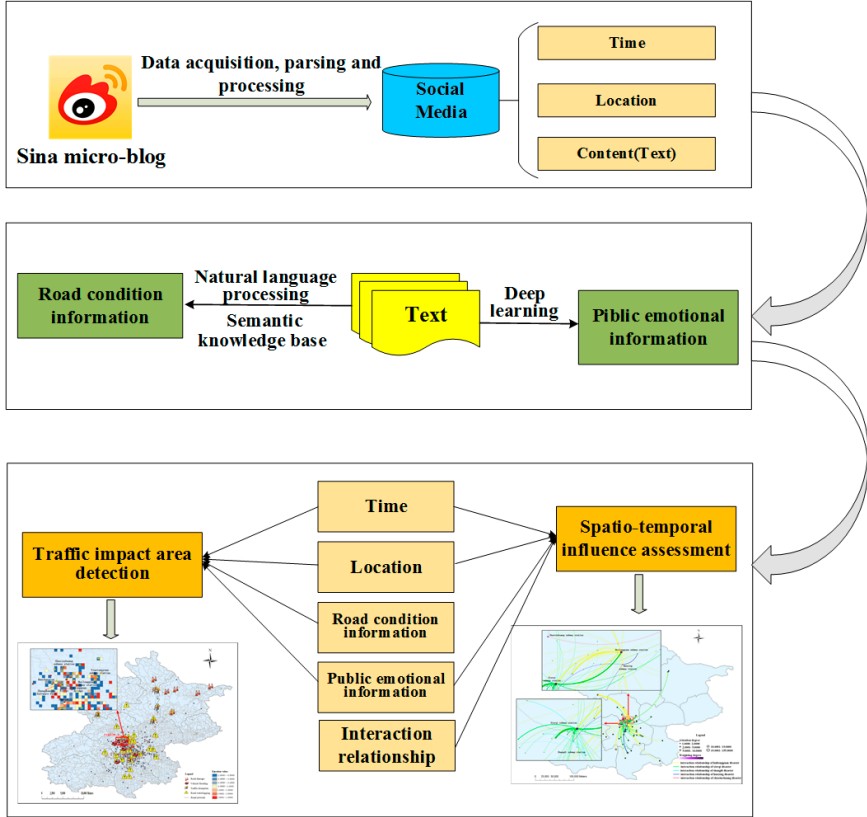

**Figure 1.** Framework of traffic impact area detection and spatiotemporal influence assessment.

### 2.1. Social Media Data Acquisition, Parsing, and Processing

We took a rainstorm disaster in Beijing, China, in 2018 as a case study to verify the role of the disaster reduction framework proposed in this paper. The main period of this rainstorm was from the night of July 15th to the night of July 17th, of which the 16th was the most serious. The storm caused serious water accumulation on many roads and even road collapse. Many cars were also damaged by standing water. The rainstorm was named the "7.16" rainstorm, which has become one of the most extreme rainstorm events in this city in the past eight years.

#### 2.1.1. Data Acquisition and Parsing

The data involved in this paper were obtained from the Sina micro-blog, which is one of the most active social media platforms in China. The platform provides API (Application Programming Interface) interfaces for the public to obtain their data. However, this method has some limitations. For example, the obtained data have no location information, and there is much irrelevant information. Thus, in this paper, we used computer programming to develop a data acquisition tool. The tool was based on the advanced search platform of the Sina micro-blog. By setting the specified parameters, such as time period, regional scope, and keywords, we obtained the desired disaster-related data. The original social media data were in the form of HTML. Through data parsing, we separated the specified information including time, location description, and text content.

In the case of this paper, we set the whole Beijing region as the study area, and the time range was from 20:00 on July 15th to 12:00 on July 18th. The keywords were set as "rainstorm", "heavy rain", and "flood", to improve the amount of retrieved data.

#### 2.1.2. Data Processing

Social media data are uploaded spontaneously by the public. These individuals usually have no professional background on disasters, and their language usage habits are also different. Therefore,

the original social media data often have the disadvantages of inconsistent editing standards and high redundancy, which need further cleaning and processing before being used. In this study, we mainly processed text data and location data.

The location data contained in social media data usually exist in the form of an address description, such as "Xierqi subway station" and "Houchangcun road". Data in this form cannot be analyzed and used directly. Thus, in this paper, we took the geocoding tool Amap (https://lbs.amap.com/api/javascript-api/guide/services/geocoder) to transform these address descriptions into latitude and longitude coordinates. Among them, for point data, we took the center point coordinates (such as "Zhongguancun Software Park"), and for line data, we took the middle point coordinates (such as "Houchangcun road").

Text data processing mainly included data consistency, abnormal data clearing, and redundant data de-duplication. Data consistency ensures that the format of text data is consistent, including converting full-width characters to half-width characters, traditional Chinese to simplified Chinese, etc. This can effectively improve the efficiency of the text mining algorithm. Abnormal data clearing is mainly used to remove the data that are not within the specified time and space. This is caused by search errors in the Sina micro-blog advanced search platform. Finally, after eliminating duplications, we obtained more than 24779 social media data with specified time, regional scope and topic.

### 2.2. Traffic Impact Information Mining from Social Media Text

In this paper, we mined disaster-related information from social media text from two different aspects, including fine-grained road condition information and public emotional information, which reflect the traffic impact from the public objective description and subjective feelings perspectives.

### 2.2.1. Fine-Grained Road Condition Information Extraction

There is much fine-grained road condition information in social media text. For example, a micro-blog said, "after a heavy rain, Houchangcun road was flooded directly, and many vehicles were also flooded". In this sentence, the phrases "Houchangcun road - flooded" and "vehicles - flooded" can clearly describe the disaster situation. Thus, we need an automated method to extract this information. However, there are two disadvantages in using the traditional supervised learning method. 1) Social media text is usually shorter and more similar to spoken language, and a text usually contains different categories of road condition information. It is easy to make the text feature sparse and semantic information fragmented. 2) The lack of an available annotation corpus makes it difficult to apply a supervised learning method, and it is time-consuming and labor-consuming to build such an annotation corpus manually. In previous studies, we realized these issues, and through considering the Chinese semantics and grammar rules, we proposed a method combining natural language processing and a semantic knowledge base of hazard damage information [27]. This method constructs a rule template of feature word collocation from the word level granularity. The template is then applied to extract candidate words that may represent hazard damage information contained in social media data. These candidate words are matched with the hazard damage semantic knowledge base to determine whether they belong to a certain hazard damage category. This knowledge base is supplemented and enriched by using third-party knowledge bases (such as Synonymy Thesaurus provided by the Harbin Institute of Technology) and a massive disaster-related corpus. Relevant experiments have shown that this method has a good performance in hazard damage information extraction in typhoon disasters. Based on the case in this paper, we used this method to extract fine-grained road condition information from social media data.

### 2.2.2. Public Emotion Information Extraction

Many researchers have studied how to extract public emotional information from social media data. Related methods can be divided into emotional dictionary-based methods [28,29] and traditional machine learning methods [30,31]. The former takes the emotion dictionary to match the emotion words in the text and then takes the weight of each emotion word to calculate the emotion

value of the whole text. This method is usually simple and effective, especially for text with strict language expression rules, such as news. However, this method is not suitable for social media data with serious colloquialism. This is because social media text often contains many network words with an emotional tendency, such as "凉凉 (It's done)", which represents negative emotion. These words are difficult to be loaded into the emotional dictionary. The latter is to train the model by the annotation corpus of different emotion categories, and then the trained model can automatically calculate the emotional tendency of text data. This method is not affected by network words. Common models include SVM (Support Vector Machine) [32], Naive Bayes [33], etc. With the development of data mining technology and high-performance computing, researchers have increasingly focused on deep learning models to extract emotional information from text [34]. Compared with traditional machine learning methods, deep learning algorithms can consider the context semantic information of text and achieve more efficient results [35]. This is because 1) traditional machine learning models are based on manual extraction of feature words (words that can represent emotional categories), and the quality of feature word extraction directly affects the classification accuracy of the model. 2) Traditional machine learning models take a bag-of-words model to handle these manually extracted feature words. This method ignores context information in text. Thus, it is difficult to use text when the feature words are not obvious or the emotion category tends to depend on context. Instead, the deep learning algorithm takes the word vector model instead of the bag-of-words model. The word vector model is obtained by training large-scale related text sets, and the trained model can transform every word in the text into a high-dimensional vector. These vectors contain rich-context semantic information. Through the iterative calculation of deep learning models, some feature words that can determine the emotional tendency of the text can be extracted automatically. These are helpful to improve the recognition accuracy and efficiency of the model. Thus, in this paper, we took a convolutional neural network to extract emotional information from text according to documents[12].

### 2.3. Detection Method of Traffic Impact Areas Based on Multi-Source Disaster-Related Data

Beijing is one of the largest cities in China with an enormous population. According to statistics, by 2018, the permanent population of Beijing had reached 21.542 million [36]. Every day, a large number of human travel activities occur in the city, which creates much pressure on traffic. When rainstorm disasters occur, urban waterlogging is very harmful to traffic conditions. Therefore, we proposed an analysis process, using the multi-dimensional disaster information extracted from social media, to analyze the spatiotemporal characteristics of the traffic impacts during rainstorms. The overall flow is shown in Figure 2.

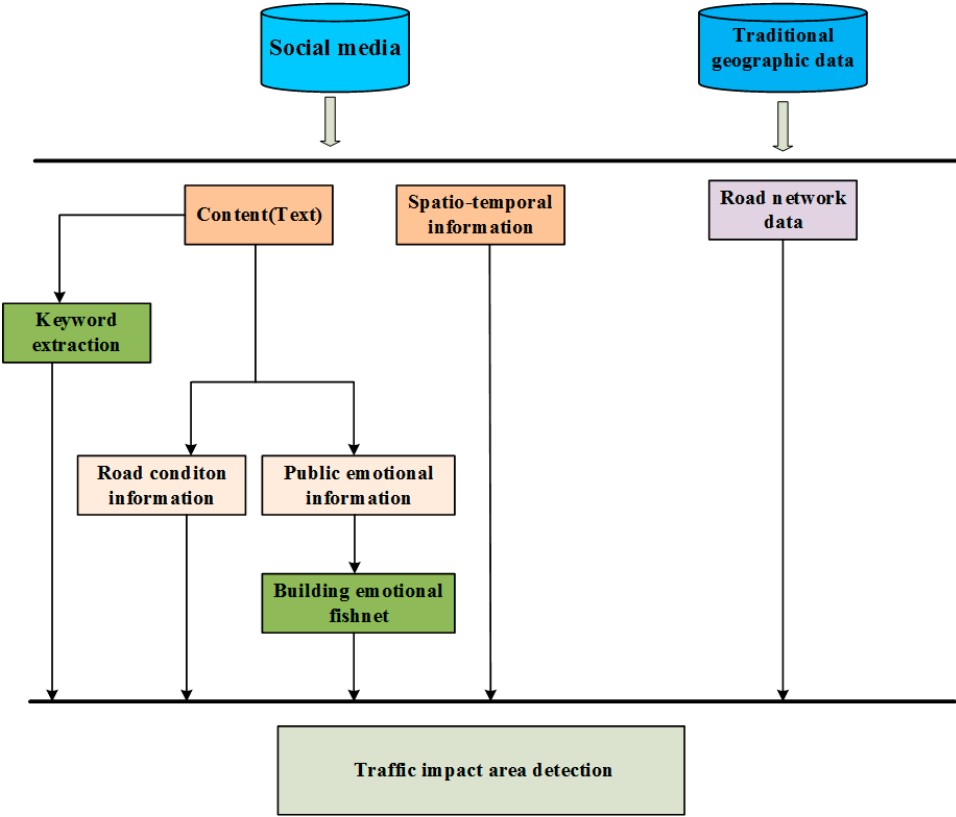

**Figure 2.** Workflow of the data processing.

(1) Data input:

The data we inputted mainly included social media data and road network data from Beijing. Among them, the social media data included the time, location coordinates, content, and extracted disaster information (public emotional information and fine-grained road condition information). The road network data of Beijing came from the "National Catalogue Service For Geographic Information" (http://www.webmap.cn/commres.do?method=dataDownload).

(2) Building an emotional fishnet:

Emotional information expressed by social media users may also reflect the emotional tendencies of their neighbors or the whole community, even if these neighbors or communities did not use social media [10]. Therefore, we need a method to allocate emotions to a certain size area around the user's location, that is, to transform the discrete point data into polygon data with certain attribute values. This paper aimed to achieve this goal by building an emotional fishnet. We divided the whole research area into some small grids. Then, we summed the emotional values of all points in each small grid, and assigned average value to these grids. The calculation formula of the average value of each grid is as follows:

$$E_j = \sum_{i=1}^{n} e_i \tag{1}$$

In this formula, $E_j$ is the emotional value of a small fishnet, $n$ represents the number of points in the fishnet, and $e_i$ represents the emotional value of each point in the fishnet. We defined positive emotion as 1, negative emotion as - 1, and neutral emotion as 0.

(3) Keyword extraction:

Emotional information is the public's subjective feelings about a disaster. It is an effective supplement to road condition information and reflects the severity of the affected areas. Keyword extraction can reflect reasons for public emotional expression because not every piece of micro-blog data contains road condition information. In this paper, we took a tool

(http://www.picdata.cn/picdata/) to extract keywords from micro-blogs that contained related emotional information.

(4) Traffic impact area detection combined with time information:

We overlaid the relevant disaster information in space and then analyzed the characteristics of the disaster in different areas of interest. This greatly improves the detection accuracy of affected affected areas.

*2.4. Construction of a Spatiotemporal Influence Assessment Model of Disasters*

The method described in section 2.3 can help to detect some seriously affected areas. However, if we can evaluate the spatiotemporal influence characteristics of these affected areas, it will help us optimize disaster reduction deployment and rescue programs. Thus, in this paper, we proposed a spatial influence assessment model by combining time, spatial location, and disaster information (public emotional information) and built an interaction diagram to assess the spatiotemporal influence of disaster in different areas.

We refer to some modeling indicators in complex networks, such as node degree, interaction degree, and weighting degree. These indicators have been applied to the field of tourism to study the popularity of scenic spots and achieved good results [37]. Among them, the node degree reflects the importance of nodes in the network, interaction degree reflects the frequency of interaction between two nodes, and weighting degree reflects the interaction degree of a node in the whole network. In this paper, we considered the co-reference relationship of events to represent interactions between nodes. For example, when a certain area $v_i$ is affected by a rainstorm and another area $v_j$ has micro-blogs talking about $v_i$, then $v_i$ and $v_j$ establish an interactive relationship. Furthermore, in the case of this paper, we optimized the indicators in [37] and proposed the attention degree, interaction degree, and weighting degree to combine emotional information and to assess the spatiotemporal impact of the affected area.

(1) Attention degree:

In the model built in this paper, we merged micro-blogs with the same geographical location into one node. The place where the disaster event occurred was used as the event node, and the place pointing to this event node was denoted the child node. For example, we took the serious water accumulation event in "Xierqi" as event node $v_i$ and the "Zhongguancun Software Park", where there were many micro-blogs discussing this water accumulation event, as the child node $v_j$. Each micro-blog in the child node ("Zhongguancun Software Park") pointed to the event node ("Xierqi"). We used indicator $D$ to indicate the attention degree of these nodes. This indicator is considered to be an attribute value of a node, which is related to the number of micro-blogs at this node discussing the specific disaster event. It indicates the impact of a designated disaster event on the area where the node is located. It is generally believed that micro-blogs with negative emotions indicate that the event had a greater influence or people paid more attention to this event. Thus, we introduced public emotional information into the calculation formula of the attention degree:

$$D_i = \sum_{k=1}^{n} E_k \tag{2}$$

Here, $D_i$ is the attention degree of every node, including the event node and its child nodes, $n$ is the number of micro-blogs contained in each node, and $E_k$ is the public emotional value of each micro-blog contained at node $v_k$.

(2) Interaction degree and weighting degree:

The interaction degree generally refers to the frequency of interaction between two nodes, which reflects the active degree of information flow between nodes. When node $v_j$ points to node $v_i$, the number of micro-blogs at node $v_j$ is the interaction degree. In the interaction diagram built from the model, the interaction degree is reflected in the thickness of the line connection between two nodes. The weighting degree reflects the sum of the frequencies to which the nodes are connected [38]. For example, the weighting degree of node $v_i$ is the sum of the interaction degrees of all nodes $v_j$ pointing to it. Based on the disaster scenario in this paper, we combined the emotional information

of micro-blogs to optimize the weighting degree calculation formula. We considered that negative emotions can increase this weighting degree. The calculation formulas of the interaction degree and weighting degree are as follows:

$$S_i = \sum_{i \in N_i} N_j \cdot E_j \tag{3}$$

$$W_{ij} = N_j \tag{4}$$

Here, $N_i$ is the set of adjacent points of node $v_i$, and $W_{ij}$ is the interaction degree between node $v_i$ and node $v_j$. If there is no interaction between node $v_i$ and node $v_j$, that is, node $v_i$ has nothing to do with node $v_j$, then $W_{ij} = 0$. $E_j$ is the emotional value of each micro-blog contained in node $v_j$, and we defined negative emotion as a value of 1.5; both positive emotion and neutral emotion took a value of 1. That is, when more nodes contain negative emotions, the weighting degree of the node ($v_i$) they point to is higher.

## 3. Results

Based on the methods in sections 2.1 and 2.2, we obtained and processed social media data related to the Beijing "7.16" rainstorm disaster and extracted the structured traffic impact information contained in it. Furthermore, combining the methods provided in sections 2.3 and 2.4, we detected the traffic impact areas, analyzed its spatiotemporal characteristics, and effectively assessed the spatiotemporal influence of related affected areas.

### 3.1. Traffic Impact Information Extraction Results

#### 3.1.1. Experimental Corpus Processing

In the experiment of fine-grained road condition information extraction, we combined social media data regarding a rainstorm in Beijing in 2018 and a typhoon passing through the mainland in 2017 [39] to expand and enrich the semantic knowledge base in this paper. We optimized the disaster damage information classification proposed in [12] and divided it into four categories in detail, according to the road conditions that often appeared in the corpus, as shown in Table 1. We labeled approximately 80 texts for each category as the test corpus, with nearly 320 texts in total. In addition, we defined that level 4 indicated the most serious impact on the road, and level 1 was the opposite. In some sentences, there may be multiple traffic impact levels, such as "暴雨过后，该处路段积水严重，造成了整条路封闭. (After the rainstorm, there was serious water accumulation in this road section, resulting in the closure of the whole road)". This sentence contained levels 1 and 2, and we took level 2 to represent the disaster situation of this text.

**Table 1.** Classification of road conditions.

| Road network traffic impact level | Category | Feature word pairs | Example sentence |
|---|---|---|---|
| Level 4 | road damage | ［公路–冲毁］([Road-Destroyed]), etc. | 密云很多**公路**都被暴雨洪水**冲毁**了. (Many **roads** in Miyun in Beijing have been **destroyed** by the rainstorm.) |
| Level 3 | vehicle flooding | ［汽车–淹没］([Car-Flooded]), etc. | 回龙观桥下，不少**汽车**被**淹没**. (Under the Huilongguan bridge, many **cars** were **flooded**.) |
| Level 2 | traffic disruption | ［路段–封闭］([Road section-Closed]), etc. | 北京市怀柔区河防口**路段封闭**！(The **road section** of |

| Level 1 | road waterlogging | ［路面−积水］ ([Road-Puddles]), etc. | Hefangkou in Huairou district is **closed.**) 雨后**路面**全是**积水**啊! (The road was filled with puddles from the rain!) |
|---|---|---|---|

In the emotion classification experiment, we divided public emotion information into three categories: positive emotion, negative emotion, and neutral emotion. Among them, negative emotion represented people's complaints about a rainstorm, hazard damage, etc. Positive emotion was mostly to express people's happiness due to avoiding some disaster event, gratitude to disaster mitigation, some joking, etc. Neutral emotion was related to an objective description and report about a disaster. We manually labeled nearly 1700 texts for each category, 1400 of which were used as model training and 300 as model testing.

### 3.1.2. Experimental Environment

The whole algorithm flow was realized using the Python language. The natural language processing tool "Hanlp" (http://www.hanlp.com/), which is an open source toolkit and can provide word segmentation and part of speech tagging functions, was used. The Word2vec model was used to calculate semantic similarity and vectorize words in text data. A convolutional neural network was built by using the TensorFlow framework [40], and we optimized the model parameters to achieve better results.

### 3.1.3. Experimental Results

We verified the accuracy of the algorithm based on the precision (P), recall (R), and comprehensive evaluation indexes (F-1). The formulas are shown below:

$$P = \frac{N\_Correct}{N\_Correct + N\_False} \tag{5}$$

$$R = \frac{N\_Correct}{N\_Category} \tag{6}$$

$$F\text{-}1 = \frac{2 \times P \times R}{P + R} \tag{7}$$

$N\_Correct$ represents the number of texts that were correctly classified into one category, $N\_False$ represents the number of texts that were misclassified into this category, and $N\_Category$ represents the number of texts that belonged to this category in the testing corpus.

Table 2 and Table 3 show the extraction accuracy of road condition information and public emotional information, respectively. Among them, the comprehensive evaluation index of different categories of road condition information is over 72%, and that of the public emotional information is over 78%. These accuracies meet the requirements of the experiment in this paper.

**Table 2.** Classification accuracy of different damage categories.

| Road condition information category | P | R | F-1 |
|---|---|---|---|
| Road damage | 90.13% | 72.50% | 80.35% |
| Vehicle flooding | 86.75% | 80.00% | 83.24% |
| Traffic disruption | 73.81% | 72.09% | 72.94% |
| Road waterlogging | 79.49% | 77.50% | 78.48% |

**Table 3.** Classification accuracy of public emotion classification.

| Emotional category | P | R | F-1 |
|---|---|---|---|
| Positive | 87.38% | 82. 97% | 85.62% |
| Neutral | 72.88% | 84. 49% | 78.26% |
| Negative | 85.37% | 76. 06% | 80.47% |

### 3.2. Detection of Urban Traffic Impact Areas

It was reported that July 16th was the most serious day during the rainstorm disaster. Therefore, we selected this day to study the impact of the rainstorm on urban traffic conditions. We visualized all micro-blog data from the 16th on a map and symbolized the data that contained different road condition categories and public emotion categories, as shown in Figure 3. The impact in region a was great during this rainstorm disaster. The main disasters in this region included "vehicle flooding", "traffic disruption", "road waterlogging", etc. These disasters were mainly concentrated in "Xierqi", "Houchangcun", "Zhongguancun Software Park", "Huilongguan", "Shangdi", etc. The emotional information here was mainly negative, indicating that the public had complained more about the rainstorm. Thus, we took this area as an example and combined time information to analyze the severely affected areas. We divided the 16th into three periods: 5:00-10:00 (morning peak), 10:00-17:00 (other periods), and 17:00-22:00 (evening peak). Furthermore, we symbolized the roads with detailed descriptions based on the traffic impact. For example, "**Shangdi seven street** was filled with puddles after the rain, please drive carefully!". The final results are shown in Figure 4.

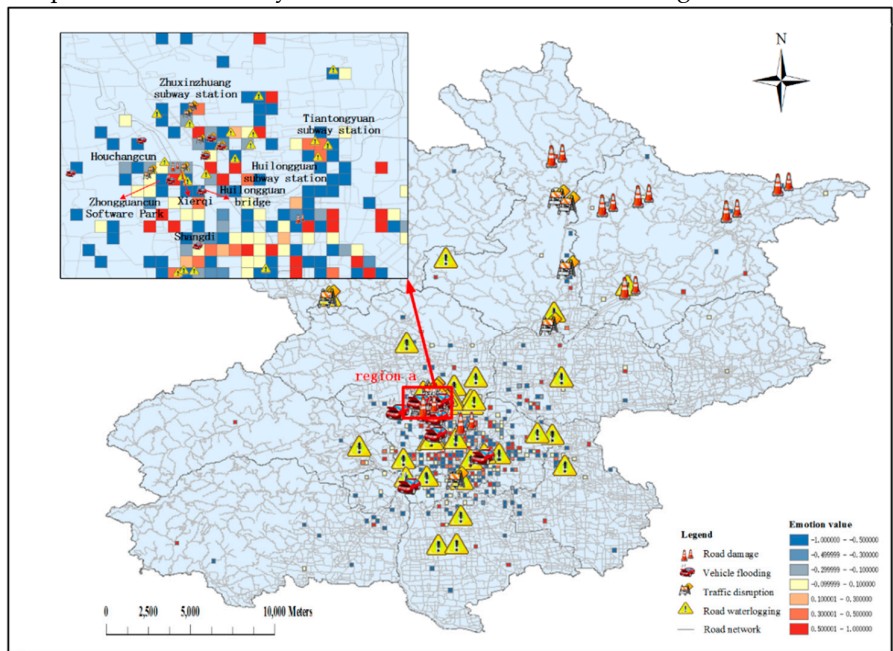

**Figure 3.** Distribution of traffic conditions in Beijing.

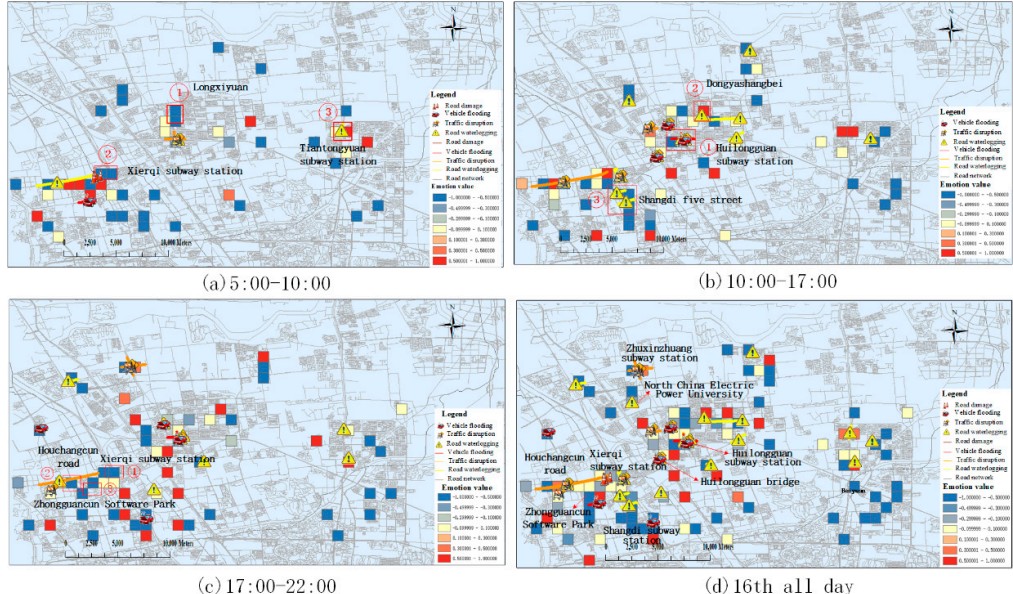

**Figure 4.** Evolution process of the disaster information distribution on the 16th. The figure (a), (b), (c) and (d) describe the disaster information in different time periods on July 16th. Among them, the time period described in figure (a) is the morning peak, figure (c) is the evening peak, figure (b) is the other periods, and figure (d) is the whole day on the 16th. The red circles in each figure are regions of interest which will be further studied in detail.

On the 16th, the rainstorm had a great impact on traffic, and it was not limited to the morning peak and evening peak. In Figure 4, we selected some regions of interest and analyzed the disaster from the extracted keywords, public emotion, road conditions and regional category. The comparison results are shown in Table 4. In Figure 4.a, region 2 and region 3 included only traffic stations, but they were affected differently. Through keywords, we further understood that the disaster in region 3 was not serious, and the main emotional category in this area was positive. In contrast, the disaster situation in region 2 was relatively serious, and people were generally worried that the current traffic situation would cause major issues in their work and life. Thus, region 2 should have received more attention for disaster reduction. In addition, although there was no obvious disaster in region 1, this region represents a residential area with a higher population. The proportion of negative emotions and the keyword information indicated that this area was also more affected by the rainstorm. As time continued, the impact of road conditions became more pronounced, as shown in Figure 4.b. However, in this time period, there were more positive emotions. We selected three small regions in this figure for further analysis. Table 4 shows the analysis results of these regions. We can see that region 1 and region 2 were located in "Huilongguan", Changping District. Among them, region 1 was located near "Huilongguan subway station". There was a large amount of water on the road, which caused many vehicles to be flooded. Most people felt sad about flooded cars and worried about travel safety. Region 2 was a residential area near "Huilongguan subway station". This region was also greatly affected by the rainstorm. The road nearby was heavily flooded. However, it can be learned from the keywords and corresponding text content that the relevant disaster situation had been properly alleviated (some disaster relief workers were helping clean up, and even some residents also joined spontaneously). The public was very satisfied with this and expressed more positive emotions. Region 3 was located near "Shangdi", and it was also heavily affected by rainfall. We can see that "Shangdi seven street" and "Shangdi five street" were flooded, and people expressed more negative emotions about the situation. During the evening peak period, although there was not much disaster information across the whole area, the public expressed more negative emotions, as shown in Figure 4.c. We selected three regions, all of which belonged to the "Xierqi" area (region 2 was closely related to region 1 and region 3). Among them, region 1 was near "Xierqi subway station"

and region 3 was "Houchangcun road". Region 2 was the nearby office zone, namely, "Zhongguancun Software Park". Every day, a large number of staff workers go to the "Zhongguancun Software Park" to work, and they travel through "Xierqi subway station" and "Houchangcun road". Therefore, although there was not much road condition information in these three regions, they all had obvious negative emotions, and the keywords of the micro-blogs in these regions were similar. People worried that they would not be able to go home on time due to the rainstorm (it was reported that at 17:00, rain began to appear in this area), especially in region 1 and region 3. Figure 4.d shows the affected conditions of the whole area on the 16th. Combining the analysis results of the other three time periods, we found that the disaster was severe near "Xierqi Subway Station" and "Huilongguan Subway Station", and the public expressed more negative emotions. There were many flooded vehicles, especially near "Huilongguan subway station". The disaster near "Shangdi subway station" was also serious. Some nearby roads were flooded, which had a great impact on traffic flow.

**Table 4.** Disaster characteristics in different regions and in different time periods.

| Region | Major Emotion Category | Road condition information | Regional category | Keyword |
|---|---|---|---|---|
| region 1 (Figure 4.a) | negative | | residential area | 暴雨 (rainstorm), 担心 (worry), 上班 (go to work), etc. |
| region 2 (Figure 4.a) | negative | traffic disruption | traffic station | 看海 (see the sea), 游泳 (swimming), 堵 (congestion), 迟到 (be late), etc. |
| region 3 (Figure 4.a) | positive | road waterlogging | traffic station | 大雨 (heavy rain), 好久不见 (long time no see), etc. |
| region 1 (Figure 4.b) | negative | vehicle flooding, road waterlogging | traffic station | 车辆 (vehicle), 淹没 (submerge), 安全 (security), etc. |
| region 2 (Figure 4.b) | positive | traffic, disruption, road waterlogging | residential area | 积水 (ponding), 清除 (eliminate), 辛苦 (toilsome), etc. |
| region 3 (Figure 4.c) | negative | road waterlogging | road | 积水 (ponding), 淹没 (flood), 通行 (traffic), etc. |
| region 1 (Figure 4.c) | negative | | traffic station | 回家 (go home), 船 (boat), 下雨 (rain), 拥堵 (congestion), etc. |
| region 2 (Figure 4.c) | negative | | office zone | 下班 (go off work), 回家 (go home), 下雨 (rain), 讨厌 (hate), etc. |
| region 3 (Figure 4.c) | positive | road waterlogging | road | 拥堵 (congestion), 回家 (go home), 下雨 (rain), etc. |

*3.3. Spatiotemporal Influence Assessment of Disasters*

It was reported that in this rainstorm disaster (on the 16th), many areas were seriously affected, especially some transportation stations, such as "Xierqi subway station", "Huilongguan subway station", and "Shangdi subway station". In these areas, road waterlogging was serious and many vehicles were flooded, which had a great impact on human travel. The analysis results in section 3.1 also showed this scenario. In this section, we selected five areas that were heavily affected by the rainstorm in Figure 4.d, including "Xierqi subway station", "Huilongguan subway station", "Huoying subway station", "Shangdi subway station", and "Zhuxinzhuang subway station", and we applied the spatial influence assessment model proposed in section 2.3.2 to further analyze them. We used an interaction diagram to visualize the analysis results of the model, as shown in Figure 5.

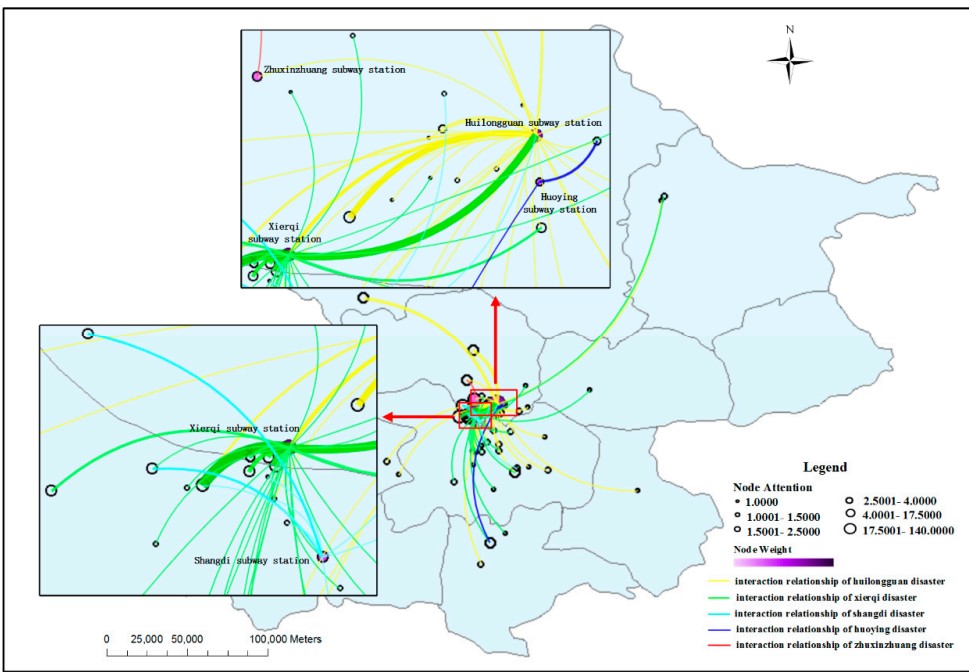

**Figure 5.** Geospatial distribution of influence for selected nodes on the 16th.

In Figure 5, the thickness of the line connecting two nodes represents their interaction degree. The thicker the line is, the more obvious the interaction between two nodes. The size of the node represents the attention degree of the node. The larger the value is, the greater the impact of the disaster on the location of the node; the attention degree of the event node indicates the local influence of the event on its location. When two nodes $v_j$ and $v_k$ are connected with an event node $v_i$ and they have the same interaction degree with this event node, the node with the high attention degree is impacted to a greater extent by the event, i.e., it contains more micro-blogs with negative emotions. In the entire network, only the five selected event nodes have a weighting degree. This is because the child nodes do not have any other node that points to them. An event node with a high weighting degree indicates that this event has greater spatial impact, and the color of this node is darker in the interaction diagram. Figure 6 shows a detailed comparison of the five selected event nodes in the attention degree and weighting degree on the 16th. We can see that disaster events at "Xierqi subway station" and "Huilongguan subway station" had high attention degrees and weighting degrees. Among them, the node at "Huilongguan subway station" had a greater impact on the local area, while the node at "Xierqi subway station" had a greater impact on other areas. This showed that the spatial influence of the disaster event at "Xierqi subway station" was more significant, and many people in other places paid more attention to the disaster situation in this area. "Zhuxinzhuang subway station" is an important transportation hub and has a large daily passenger flow. However, the attention degree and weighting degree of this node were not large. This showed that the disaster situation at this node was not serious and had a limited influence. The disaster situation around "Huoying subway station" was similar to that around "Zhuxinzhuang subway station". Although

the attention degree and weighting degree of the node at "Shangdi subway station" were far less than those of the nodes at "Xierqi subway station" and "Huilongguan subway station", the weighting degree was far greater than the attention degree. This result showed that the spatial influence of the disaster at this node was large.

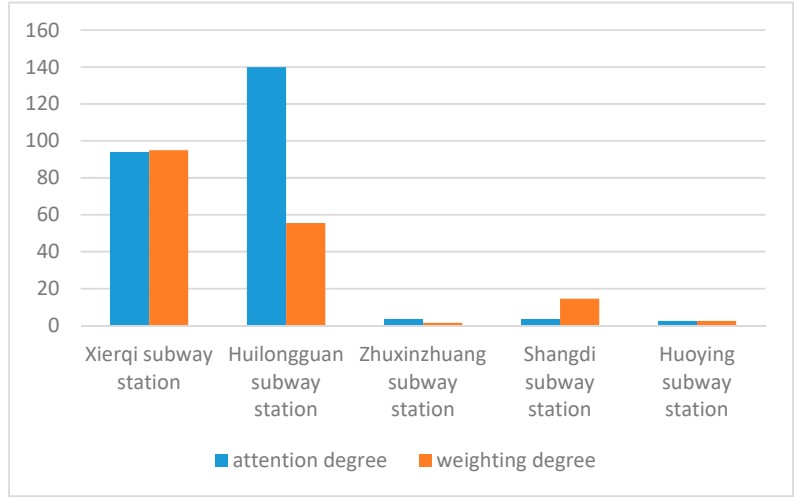

**Figure 6.** Node degree comparison of selected nodes on the 16th.

Based on the results of the above analysis, we selected two nodes located at "Xierqi subway station" and "Huilongguan subway station" as the research objects for further detailed analysis. We added the variable 'time' to study the spatiotemporal influence of these two nodes. We selected the 16th, when the rainstorm was the worst, and divided it into three periods according to section 3.1. The final result is shown in Figure 7.

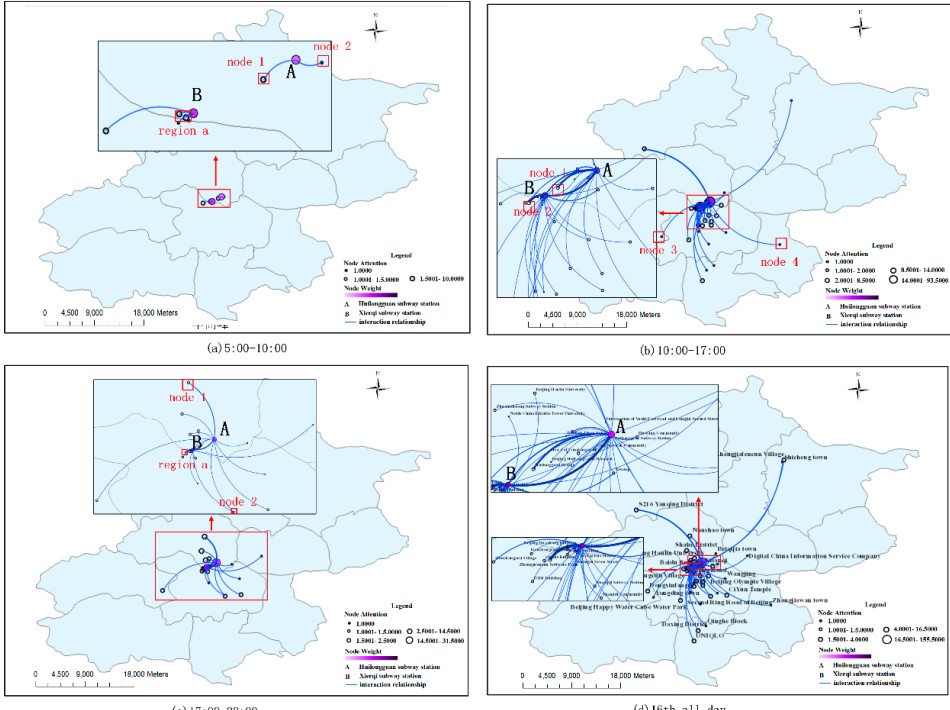

**Figure 7.** Changes in the spatiotemporal influence of the disaster situation at "Xierqi subway station" and "Huilongguan subway station". The figure (a), (b), (c) and (d) describe the magnitude and distribution of spatial influence of these two nodes in different time periods. Among them, the time

period described in figure (a) is the morning peak, figure (c) is the evening peak, figure (b) is the other periods, and figure (d) is the whole day on the 16th. The red rectangles in each figure are regions of interest which will be further studied in detail.

In Figure 7.a, we can see that there were very few other nodes connected to node A (representing "Huilongguan subway station"). We selected two nodes (node 1 and node 2) connected to node A for detailed analysis. Node 1 and node 2 had the same interaction degree with node A, i.e., the thickness of the connecting lines between them were the same. However, the attention degree of node 1 was larger, which showed that node A had a greater impact on node 1. We checked the corresponding micro-blog content to confirm this and found that people were worried that the disaster situation at node A would affect their travel. Compared with node A, node B had many external nodes connected to it, such as region a in Figure 7.a. We found that region a was mainly an office zone ("Zhongguancun Software Park"), where many enterprises were located, such as "Baidu" and "Lenovo". The attention degree of the nodes in this region was relatively large. By checking the corresponding micro-blogs, we found that the public mostly expressed their aversion to disaster at node B because it affected them going to work normally, and they were also worried that their colleagues would not pass through this area. Figure 7.b shows that the time period was from 10:00 to 17:00. In this period, the main influence areas of node A and node B were quite different from that shown in Figure 7.a. The effects began to spread to the whole city, especially in the south (city proper). In other words, the impact of these two nodes began to expand from local impacts to external effects. We selected four child nodes, including node 1, node 2, node 3, and node 4, to analyze. Node 1 and node 2 are located near node A and node B, and the corresponding nodes had larger attention degrees and interaction degrees. This result showed that people were very concerned about disasters that occurred at nodes A and B, especially at node 1, which was located around "Huilongguan bridge". We checked the corresponding micro-blogs and found that node 1 was also greatly affected by the rainstorm event and that the disasters at nodes 1 and A had caused great inconvenience to the travel of local residents. The interaction degrees and attention degrees of node 3 and node 4 were not large, but they were far away from event nodes A and B. This result indicated that node A and node B had a greater spatial influence in this period. In the evening peak period, which was from 17:00-22:00, as shown in Figure 7.c, we can see that the number of nodes pointing to node A and node B was lower than that of Figure 7.b, but they spread more evenly around the area, showing that there was a wide range of attention to these two nodes. We selected some interesting regions and nodes, including region a, node 1, and node 2. Among them, region a was located in "Zhongguancun Software Park". People at this node were worried that they would not be able to go home on time because nodes A and B were affected by the rainstorm. Node 1 was located in "Nanshao town", and people there worried about the rainstorm hitting their homes in the future when they saw the severe disaster at node A and node B. However, the disaster reduction department could provide timely and effective reassurance to avoid the intrusion of false information. Node 2 was located in the inner city of Beijing, and there had also been serious flooding incidents here. People mentioned the disaster situation at nodes A and B at the same time to express their dissatisfaction with urban drainage. Figure 7.d shows the spatial extent and distribution of the disaster impact of node A and node B throughout the entire day of the 16th. We can see that during this rainstorm disaster, the spatial influence of the disasters at node A and node B was relatively large, especially for the surrounding residential area and office zone.

## 4. Discussion

### 4.1. Discussion on Extraction Accuracy of Disaster-Related Information

Fine-grained road condition information extraction plays an important role in effectively detecting and analyzing traffic impact areas. The relevant extraction results provide powerful data support for targeted disaster relief. However, the shortcomings of semantic fragmentation and discrete feature of this information make many traditional machine learning methods difficult to

apply. Although the extraction accuracy of the method in this paper meets the requirements of the experiment, this method still has room for optimization. We carefully analyzed some misclassified text data and summarized the following two improvements: 1) some sentences, such as "滴滴大厦附近被水淹了 (flooded near Didi Building)", cannot be effectively identified. This is because we cannot exhaustively list all geographic-related named entities, such as "滴滴大厦（Didi Building）". Thus, a method that can automatically recognize these words[41] should be introduced to improve the algorithm recognition efficiency in the future. 2) This method cannot process some text containing implicit semantic information, such as "我在西二旗看海 (I watch the sea at Xierqi)". The word "看海 (watch the sea)" represents that the road is flooded in this context. Thus, we should consider a more relevant corpus and use clustering algorithms, such as an LDA (latent Dirichlet allocation) model [42], to obtain similar words to supplement our disaster knowledge base.

Public emotions can be regarded as an effective supplement information to fine-grained road conditions and help with the detection and analysis of traffic impact areas. In this paper, by analyzing the characteristics of disaster-related social media text, we used a convolutional neural network model to extract this information. The final accuracy meets the requirements of the experiment. However, some indicators still need to be improved, especially the precision of neutral emotion and the recall of negative emotion. We analyzed misclassified text data and found that a possible major factor is the number of similar texts that belonged to different emotional categories, and they were distinguished by some difficult-to-abstract feature words, such as punctuation marks and emojis, e.g., the text "Beijing, heavy rain" and "Beijing, heavy rain😭 ... ". The two texts have similar structures, but the former belongs to the neutral category and the latter belongs to the negative category. Thus, in our next study, we will consider introducing artificial feature engineering to improve the classification ability of the model, such as extracting various categories of emojis from the training text data and labeling them with emotional weights to guide the model for classification calculations.

### 4.2. Discussion on the Results of Disaster Spatiotemporal Analysis

In section 3.2, we combined different disaster-related information, including time, location, road conditions, public emotion, road network, etc., to study the role of social media in detecting traffic impact areas. These information sources complemented each other and helped us better understand the disaster situation. Based on the results in section 3.2, we obtained the following knowledge: 1) in the study area shown in Figure 4, we can see that the transportation station areas were more seriously affected, such as "Huilongguan subway station", "Xierqi subway station", and "Shangdi subway station". This is because there was not only more disaster-related road condition information in these places but also more negative emotions, which reflected the severity of the disaster. In addition, some residential and office areas near these transportation stations were also greatly affected, such as "Zhongguancun Software Park" and "Dongyashangbei". These areas contained more negative emotions, especially during the morning and evening peaks. This is because during these time periods, people typically have more travel activities, and bad traffic conditions affect them more. Thus, some disaster prevention work designed to ensure smooth traffic should be designed according to the spatiotemporal distribution characteristics of relevant disaster situations, especially for areas with substantial traffic impact information. During the time period described in Figure 4.b, some "vehicle flooding" appeared in the study area. These disasters were mainly located near "Huilongguan Subway Station" and "Huilongguan bridge", which may have been caused by some drivers incorrectly estimating the water depth. When such disaster events occur, the disaster reduction department can launch rescue efforts in a timely manner and release relevant disaster information through various channels, including social media, to prevent more accidents, which can greatly improve the efficiency of disaster reduction and rescue. 2) Public emotion, as an important supplementary information, is very important for disaster reduction. In Figure 4, many areas did not contain specific road condition information, especially some residential areas, such as region 1 in Figure 4.a. These areas were usually densely populated and had more negative emotions. They should also receive more attention as the rainstorm affected more than just public travel. Furthermore,

some psychological studies have shown that negative emotion may make people vulnerable to the induction and deception of bad information, such as rumors, during disasters [43]. For example, a rumor suggested that the residents in Beijing should stay off work because of the unprecedented rainstorm (http://www.cma.gov.cn/2011xwzx/2011xqxxw/2011xqxyw/201706/t20170623_426973.html). This would have a large impact on those people who were worried and anxious about being late for work, and they may select some unfortunate measures to eliminate this contradictory psychology[44], such as spread these rumors or easily believe other rumors. This greatly increases the risk of secondary disasters. Therefore, some necessary disaster mitigation measures should be launched in time, such as clarifying these rumors and publishing the disaster situation in a timely manner. In general, the comprehensive analysis of multi-dimensional disaster-related information gives us a more comprehensive understanding of the progress of disasters, which is of great significance to improve the efficiency of disaster reduction.

In section 3.3, we evaluated the spatiotemporal influence of different disaster-affected areas by using improved traditional complex network modeling methods. Figures 5 and 6 show the spatial influence of these areas by visualizing some quantitative indicators, including attention degree, interaction degree and weighting degree. We can see that "Xierqi subway station" and "Huilongguan subway station" were most affected by the disaster. This information is a valid supplement to the analysis results in section 3.2, and it further demonstrates the scope of the impact of these disaster areas in more detail. A more detailed analysis for the two selected areas ("Xierqi subway station" and "Huilongguan subway station") was conducted from the time dimension. Figure 7 presents the results of 3 time periods (morning peak period, other periods and evening peak period), and we found that 1) during the morning and evening peak periods, these two nodes (the two selected areas) had a greater impact on the nodes (other areas) around them, especially residential areas and office zones. Moreover, the node degree and interaction degree indicators showed how affected these areas were. Although the two selected nodes also had an effect on distant nodes, this influence was limited. Combined with the micro-blog content, we could reasonably allocate rescue resources for these affected areas. 2) During other periods, the areas affected by the two selected nodes increased. However, the magnitude of influence was not regular. We focused on areas with many nodes pointing to them. These areas may be severely affected by nodes A ("Huilongguan subway station") and B ("Xierqi subway station"). Disaster reduction departments could efficiently formulate some response plans by including micro-blog content in these areas, such as pushing current traffic conditions and recommending reasonable travel routes. Overall, the model proposed in this study was effective at assessing the spatiotemporal influence of disasters. Visual analysis results provided an important information reference to optimize disaster reduction decisions and efficiently deploy relief resources.

## 5. Conclusions and Future Work

Social media has played an important role in the research of disasters, such as urban rainstorms, in recent years. Abundant disaster-related information contained in it provides important data support for disaster reduction. However, this disaster-related information often exists in an unstructured form, which makes the information difficult to use efficiently. Thus, in this paper, we constructed a framework, which integrated algorithms including natural language processing and deep learning, to extract these multi-dimensional disaster-related information, including time, location, fine-grained road condition information, and public emotional information. Furthermore, we comprehensively analyzed this extracted disaster-related information to detect the traffic impact areas and achieved good results. In addition, based on the rich interaction patterns in social media, we proposed a spatial influence assessment model. Through visualizing several quantitative indicators, we could learn more about the magnitude and distribution of the spatial influence of different affected areas, which was of great help in optimizing disaster reduction decisions. The rainstorm disaster in Beijing on 16 July, 2018, was used as a case study to verify the effectiveness of the proposed method in this paper.

In general, the framework in this paper performed well in traffic impact area detection and the spatiotemporal influence assessment of the selected disaster. However, in future work, there are still some aspects that require further improvement. First, we will consider extracting geographic-related named entity words from social media text. On the one hand, this can improve the recognition accuracy of fine-grained road condition information (which was mentioned in section 3.1). On the other hand, not all social media data contain location information. When posting disaster-related data, people may not upload their current location due to personal habits and other reasons. This greatly limits the data usage because most disaster analysis methods rely on location information. However, some studies [45-47] have found that the geographic-related named entity words contained in social media text can indicate where data were uploaded. Second, the framework in this paper considered using traditional geographic data, such as road networks, to assist with the analysis. These data provided an effective supplement to social media and achieved good analysis results. Therefore, in our next study, more relevant data will be introduced. For example, population distribution data could help understand the regional impact of disasters, and bus route data could assist in analyzing the impact on people's travel.

**Author Contributions:** Tengfei Yang, Jibo Xie, and Guoqing Li conceived and designed the study; Tengfei Yang and Jibo Xie wrote the paper; Tengfei Yang and Naixia Mou designed and implemented the algorithmic framework; Tengfei Yang, Cuiju Chen, and Zhan Liu realized the visualization; and Zhenyu Lin and Jing Zhao collected and processed the data. All authors have read and agreed to the published version of the manuscript.

**Funding:** This research was funded by the National Key R&D Program of China, grant number 2016YFB0501504 and the Strategic Priority Research Program of CAS (XDA19020201).

**Conflicts of Interest**: The authors declare no conflict of interest.

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
