# Peer review of "Traffic Impact Area Detection and Spatiotemporal Influence Assessment for Disaster Reduction Based on Social Media: A Case Study of the 2018 Beijing Rainstorm"

_ijgi, doi:10.3390/ijgi9020136_

Round 1

Reviewer 1 Report

The paper presents a computational framework integrating several data sources (road conditions and emotional information coming from social media) to detect and analyze traffic impact during rainfall disaster. The framework has been tested on the case of the City of Beijing.

The paper is relevant for the community of the interested Journal (geo-information), allowing to enlarge the discussion towards a new generation of supporting technologies.

However, I recommend:

a better structure of the introduction (sub-sections?); a more detailed extension of the state of the art quoting not only Chinese authors; a complete revision of English language; the extension of the concluding remarks, detailing next steps of the research and future improvements of the proposed framework.

I propose to accept the paper only after the rewriting of it, following the given suggestions

Author Response

Thank you for your valuable suggestions. We have made the following changes: 1) We further divided the introduction and introduced subsections. These subchapters mainly descript the related work and the algorithm in this paper in detail. 2) We have introduced other relevant references, especially the research content of foreign counterparts. This really makes the manuscript more comprehensive and scientific. 3) We used a professional English editing company to modify the manuscript. 4) We have reorganized the conclusion part of this manuscript and put forward clear ideas for some aspects that we considers to be more important. In addition, based on the reviewers, we have adjusted the structure of the manuscript, which includes: Introduction, Methodology, Results and Discussion and Conclusion and Future Work. This makes the manuscript look clearer. Thanks again for your review.

Reviewer 2 Report

The manuscript concentrates on scrutinizing adequate and novel topic. The paper may be regarded to be as acceptable, notwithstanding, I have an advice to the author(s). The paper applies several times the notion ‘use’ in the line 23, 27, 49, 58, 62, 64, 66, 72, 73, 77, 86, 128, 131, 148, 173, 179, 185, 190, 191, 206, 209, 214, 221, 234, 235,, 250, 268, 399, 550 as well. Synonym of the word of ‘use’ ought to be found and exchanged so that the manuscript could be more sophisticated.

Author Response

Thank you for your approval of this manuscript, and for your valuable suggestions. We think this suggestion is very good. We have adopted them and revised the manuscript. At the same time, we also asked a professional English editing company to help revise this manuscript.

Thanks again for your review.

Reviewer 3 Report

The paper is well written, however it should be better organized. First of all, I advise the authors to insert a "Methodology" section which describes the various aspects of the methodologies used, better specifying the references to the literature. Furthermore, taking into account the various international frameworks on disaster risk reduction, I would recommend the authors to expand the literature by inserting references to the relationship between social media and disasters (UNIDR). Finally, I recommend that you distinguish - Introduction, Methodology, Results and Discussion - and broaden your conclusions with respect to the discussion and results.

Author Response

Thank you for your valuable suggestions on this manuscript. We have reorganized the structure of this manuscript. We have introduced five sections including Introduction, Methodology, Results and Discussion and Conclusion and Future Work. We have reorganized and compared the methods in this paper and other related references. In addition, we have also introduced some other literature to testify the important role social media plays in disaster reduction, including the report from UNISDR, VGI, etc. We have reorganized the conclusion section, including summarizing the work of this manuscript, and clarifying the next research work through the results and discussion section.

Thanks again for your review.